# A Novel Oral Syringe for Dosing and Administration of Multiparticulate Formulations: Acceptability Study in Preschool and School Children

**DOI:** 10.3390/pharmaceutics12090806

**Published:** 2020-08-25

**Authors:** Justyna Katarzyna Hofmanová, Joanne Bennett, Alastair Coupe, Jeremy A. Bartlett, Andrew Monahan, Hannah Katharine Batchelor

**Affiliations:** 1School of Pharmacy, Institute of Clinical Sciences, University of Birmingham, Birmingham B15 2TT, UK; justyna.hofmanova@gmail.com; 2Pfizer Global R&D, Sandwich, Kent CT13 9NJ, UK; Joanne.Bennett@pfizer.com (J.B.); alastair.coupe@pfizer.com (A.C.); 3Pfizer Global R&D, Groton, CT 06340, USA; jeremy.a.bartlett@pfizer.com (J.A.B.); Andrew.P.Monahan@pfizer.com (A.M.); 4Strathclyde Institute of Pharmacy and Biomedical Sciences, University of Strathclyde, 161 Cathedral Street, Glasgow G4 0RE, UK

**Keywords:** multiparticulates, administration device, acceptability, paediatric formulation, mouthfeel, oral syringe

## Abstract

The popularity of multiparticulate formulations (MPs) as a paediatric dosage form continues to increase. MPs comprise of multiple small units that are easy-to-swallow. Currently, MPs are commonly manufactured into unit doses that are either swallowed whole or opened prior to administration. While this is an acceptable approach, dosing is envisioned to be optimised with a “standard” paediatric device which can better harness the flexible dosing potential of MPs. We evaluated a novel oral syringe (Sympfiny^TM^, HS Design, Morristown, NJ, USA) that is being developed as a tool to dispense and administer MPs to children. Forty children, 4–12 years old, received 0.5, 1.2, and 2.0 mL doses of placebo MPs using the oral syringe with spring water or a drink of choice to complete sample intake. Acceptability was recorded as those able to completely swallow the dose and participants also rated dose acceptability on a 5-point scale. The ability to completely swallow the dose decreased as dose volume increased; the smallest dose was completely swallowed by 87.5% (35/40) children, and 69.4% (27/39) of children confirmed their willingness to take the sample as a daily medicine. Larger doses, 1.2 and 2.0 mL, gave values of 55% and 57.5% for the doses completely swallowed and 58.8% and 51.72% for willingness to take the sample as a daily medicine, respectively. Use of a drink of choice showed no increase in swallowability as compared with water. The novel oral syringe being developed is an appropriate device for dispensing doses flexibly and administering neutral tasting MPs directly to the mouth in the lower dose range without the need for a co-administration vehicle in children aged 4–12 years.

## 1. Introduction

A paediatric population has distinct requirements for oral drug delivery compared to adults due to their continuing development. Children present different anatomical, physiological and psychological characteristics which influence the approach to the administration of medicines [1]. Conventional formulations are not always appropriate for this patient group leading to dose manipulation and compounding of existing products [2]. Thus, an oral drug formulation specifically developed to provide dose flexibility to suit the dosing requirements across all age groups is desired.

Multiparticulate (MP) dosage forms, such as sprinkles, pellets and granules, comprise multiple units (typically below 2.5 mm [3]). These were introduced in the 1940 s and now more than 65 products are available in the US for paediatric populations [4]. MPs can be coated to provide the desired release profile or to achieve taste masking [5], which is known to improve patient compliance. Due to their small size MPs provide many advantages over single-unit systems. Their small size dramatically improves their ease of swallowing over single unit tablets, plus multi-unit composition allows flexible dosing [6]. Since each unit contains a small amount of drug, the recommended dose can be prepared by means of a counting device (such as in the case of minitablets) or measuring a given weight/volume. For these reasons, MPs offer a viable alternative to conventional liquid formulations for children [1].

The acceptability of a dosage form, defined as “an overall ability of the patient and caregiver to use a medicinal product as intended” [7], is determined by the attributes of the product and the user. For MPs, the acceptability characteristics of the formulation include taste [7], particle size, volume and mouthfeel [8,9]. The ease of administration is also vital from the user’s point of view; this includes consideration of the number of units, dosing device and (where relevant) the co-administered vehicle. The choice of administration device is an integral part of the overall quality and acceptability of the medication [10]. Existing MP medicines are typically provided as prefilled unit dose capsules or sachets where the appropriate dose is pre-packaged. The capsule/sachet may be opened to allow the MPs to be dosed to the patient directly onto the tongue [11], or given on a spoon [12]. MPs are often mixed with soft food, with applesauce being the most widespread vehicle used, to aid administration [4]. However, this requires additional testing to confirm chemical compatibility with the product and the suitability of vehicle in the target population [13] as well as the global availability of the vehicle being used. Administration of MPs with soft food is not always viable, for example, due to absence of an appropriate vehicle in which the drug product is stable or no availability of an appropriate vehicle in some parts of the world. This highlights that there is a need for an alternative, simpler administration mode.

It was demonstrated that MPs are acceptable to children when dosed on a spoon with water as the co-administered vehicle [14], yet a medicine spoon does not allow flexibility in dosing that may be required for a paediatric MP product. A prototype of a novel oral syringe (Sympfiny™, HS Design, Morristown, NJ, USA) has been designed to dispense and administer MPs directly into a patient’s mouth. This option provides enhanced flexibility in dosing from multidose containers, and may reduce costs, as compared with single-dose preparations [15].

Regulatory agencies in Europe and the United States [1,16] recommend confirming acceptability of a product in the target population, yet, without a clear guideline on the methodology to be used. In current practice, various methods and measures are used to assess acceptability of medicines [17]. When children are used as subjects, the assessment tool should be simple and easy for the children/caregivers to understand. For instance, simple Likert-type scales and hedonic scales have been often employed to evaluate specific sample attributes and/or overall acceptability of medicines in paediatric populations [18,19] as they are appropriate to use from the age of four years [20]. In addition, the proportion of participants able to swallow a dose has been recently used as a simple measure of patient acceptability in various placebo studies [14,21,22,23]. A combination of tools can minimise the bias of using a single method to report acceptability [17].

This study aimed to evaluate the acceptability of placebo multiparticulates (MPs) administered using the oral syringe Sympfiny™ in healthy child volunteers. The effect of the quantity as determined by the volume of MPs dosed on acceptability was investigated. Secondary objectives included understanding if the drink provided at the time of administration improved acceptability. Data collected as participant reported outcomes (PRO) was complemented by researcher reported outcomes (RRO).

## 2. Materials and Methods

### 2.1. Materials

Mannitol 350 mg/g placebo microsphere cores were used as model placebo MPs. The MPs contained stearyl alcohol (carrier wax), poloxamer 407 (pore former) and mannitol. They were manufactured using the melt spray congeal spinning disc atomisation technology (Lo., J.B., 2009). MPs were coated with the reverse enteric barrier membrane Kollicoat Smartseal 30 D at a coating of 10% weight gain, after coating MPs the batch was blended with 2% talc to help mitigate any tackiness or agglomeration upon storage. MPs were produced and coated under good manufacturing practices by Pfizer (Groton, CT, USA). The final MPs were passed through a screen set-up to retain cores 150–425 µm in size and displayed a monodisperse size distribution centered at approximate 246 µm (D50) in size.

A prototype 2-mL oral syringe (Figure 1) was used to dose, dispense and administer MPs. Sympfiny™ is an oral syringe and bottle adapter combination which uses a valve system to control the flow of MPs. When the oral syringe is connected to the bottle adapter, both valves open to allow MPs to flow into the syringe. Upon removal, both valves close stopping the flow of MPs. Please refer to the Appendix A (instructions for use) and the website for additional detail https://hs-design.com/sympfiny/.

### 2.2. Study Design

A cross-over, open-label, partially randomised, single-centre study design was used to evaluate the acceptability of MPs administered using an oral syringe. The study was approved by the University of Birmingham Research Ethics Committee (ERN_18-1870, 24 June 2019).

The study involved children from 4 to 12 years old; exclusion criteria were as follows: children with swallowing difficulties, children with allergies to peanuts, gluten and/or lactose. Participants were recruited via associated groups and networks from the research team via advertisements placed on relevant noticeboards and within University newsletters. Additionally, some participants were recruited at the study site where visitors expressed an interest in the study.

### 2.3. Procedure

The study took place in a designated room at Thinktank Science Museum (Birmingham, UK). In advance of the study, each participant or their legal guardian was provided with a copy of the participant information sheet. Before the study began, adult participants gave written informed consent for the study for the child in their care, and child participants gave verbal assent.

Parents/carers were asked to prepare and administer four doses of MPs, one by one, to the child in their care; self-administration (by the child) was permitted. Participants were provided with the oral syringe, one bottle containing MPs, drinking spring water, a cup that could be used to expectorate the sample, crackers, and a PRO questionnaire (Figure 2). Adult participants were given written instructions on how to use the oral syringe (Appendix A), with no additional verbal training during the study. The doses were administered in the following order: 0.5 mL, 1.2 mL and 2.0 mL, correlated to a mass of MPs of 0.32, 0.78 and 1.29 g, respectively, with water followed by 2.0 or 1.2 mL with a drink of choice (for the fourth dose, the volume was reduced during the study from 2 mL to 1.2 mL to ease the burden on the paediatric participants; this change occurred from participant #16 onwards). Participants had free access to drinking water (bottled spring water) as required to complete sample intake. For the last dose, a drink of choice—milk, squash (fruit cordial diluted with water) or fruit juice were provided as an alternative to water. Before each dose, children were recommended to use a palate cleanser—room temperature drinking water (bottled spring water), followed by a piece of lightly salted cracker (Jacob’s, or Schar gluten-free) [24]. Prior to each dose, the child was asked if they were willing to continue with the study; the adults or the children in their care were able to withdraw from the study at any point.

### 2.4. Data Collection

#### 2.4.1. Participant Reported Outcomes

Participant reported outcomes were collected using a paper-based structured questionnaire that was completed by the children immediately after each administered dose. Participants could also provide a voluntary written description of the sample, which was used to facilitate interpretation of results.

Children evaluated the samples for four palatability attributes (grittiness, volume, mouthfeel and taste) using 5-point hedonic facial scales (Table 1), replicating the previous methodology from a similar study of Lopez, Mistry, Batchelor, Bennett, Coupe, Ernest, Orlu and Tuleu [14]. After completion of hedonic ratings, participants answered the following question: “If this was a medicine and you were sick, would you be willing to take this every day?”. In addition, the number of children who reported that they felt “bits” of MPs in their mouth following administration was established (self-reported, the mouth was not inspected after swallowing). The total volume of water consumed for each sample was calculated as the difference in the mass of the cup of water before and after sample intake (ρ_H2O_ ≈ 1 g/mL).

#### 2.4.2. Researcher Reported Outcomes

Each participant was observed by a researcher during the administration of MPs. The following outcomes for each dose were reported: “completely swallowed”, “partially swallowed”, “chewed on”, “sample spat out”, “sample got stuck in throat”, “refused to take sample”, similar to previous studies [14,22,23]. During and after administration of the dose, the researcher recorded specific negative facial expressions displayed by the child (multiple responses from the following were recorded: “lips pressed (together)”, “nose wrinkling”, “eyes squeeze”, “brows pulled together and lowered”, “head shake”, “voice disgust”). The sum of negative facial expressions was calculated as an indicator of participants’ hedonic response indicating their discomfort. In addition, spontaneous verbal judgement of the samples was recorded on researcher observation sheets.

### 2.5. Data Analysis

Data from researcher observations and child participant-reported outcomes were treated as categorical data. Childrens’ ratings on a 5-point hedonic scale were assigned scores from 1 to 5 (where a score of 1 referred to a positive quality; and 5 to negative quality). Differences between scores given for two different doses (pairwise) were assessed using Wilcoxon’s signed rank test with Bonferroni correction. For comparison of multiple doses, a Friedman test was used.

Hedonic scores were used as an indicator of acceptability. Criteria to demonstrate acceptability was defined as a hedonic score ≤3 (neutral or positive face) [25]. Acceptable/unacceptable hedonic scores were then tested for association with other acceptability measures using Wilcoxon rank test.

The association between demographic data and outcome measures was examined using the Pearson Chi2 test (χ^2^). To study age-related differences, three age groups were compared: 4–6, 7–9, and 10–12 years old. The volume of water consumed was treated as a non-normally distributed continuous variable (based on Shapiro–Wilk test).

For all tests the level of significance *p* < 0.05 was used unless stated otherwise. Statistical analysis was performed using SPSS statistical software version 26 (IBM Corp, Armonk, NY, USA).

## 3. Results

The study included 40 paediatric participants. There were ten cases of discontinuation based on a child’s withdrawal, which occurred at different times of the study. The data collected up to the point of participant withdrawal were included in the analysis. Thirty children completed the full course of the study. Participants’ demographics are given in Table 2. Within the study population, 87.5% (35/40) of children had experience of taking oral medicines from an oral syringe.

### 3.1. Success of Swallowing the MPs Dose

The majority of children managed to swallow the dose of MPs (as opposed to those who partially swallowed the sample, spat out or refused it). The success of swallowing depended on the dose volume (Chi2, *p* < 0.05); the first dose administered, 0.5 mL (0.32 g), was completely swallowed by 87.5% (35/40) children (Table 3). As a larger dose volume was presented to the child, the proportion of children who only partially swallowed the sample increased, suggesting difficulty in swallowing large volumes of MPs in a single administration/mouthful. Likewise, the number of children who refused to take a dose increased with dose volume. While only one child refused the 0.5-mL dose, the larger doses were refused by 11 children (27.5%). The age of the children that refused the sample ranged from 4 to 8 years old. The children who spat out the dose tended to refuse the next dose (5 out of 7 occasions).

For the fourth dose given, children chose the drink taken with the sample from a selection of milk, fruit juice or fruit squash. It was assumed that a drink, other than water, could improve the acceptance and hedonic response to the MPs. However, no improvement in the success of swallowing the dose, nor palatability was found for the doses given with a drink of choice (dose–volume matched, Wilcoxon Test *p* > 0.05).

### 3.2. Participants Hedonic Response

Children evaluated grittiness, sample volume, mouthfeel, and taste of the doses using 5-point hedonic facial scales (Figure 3). Grittiness perception received high scores i.e., scores of 4 and 5 (where 5 was very gritty) by 47% of children. On the contrary, other palatability parameters received positive to neutral ratings (i.e., scores 1, 2 and 3) in 70%, 67% and 63% of evaluations of volume, mouthfeel and taste, respectively. No association was found between the dose size and the rating of the sample taste or mouthfeel, including the grittiness (Wilcoxon’s test, *p* > 0.05). However, based on the sample volume, the children preferred the smallest dose of 0.5 mL (Wilcoxon’s test, *p* < 0.01, Figure 3).

Participants’ voluntary feedback supported some of the findings on the hedonic scales. The taste of the MPs was described as “plain” or “tasteless”; however, the feeling in the mouth was found to be “very gritty”, “tickly” and “crunchy”. Moreover, for the larger doses (1.2 mL and 2.0 mL), the volume of the dose was often criticised by children—“it was a lot”, “too much”, and also by their parents “the dose is too big for his tongue”. This explained why the smallest dose received the most positive hedonic ratings.

The effect that an alternative drink, to chase the MPs dose, had on palatability was evaluated. Based on hedonic scales, no difference between samples taken with water or a drink of choice was found (Wilcoxon’s test, *p* > 0.05, Figure 3). Children’s anecdotal comments were also contradictory on that matter, with some describing improvement of taste with a drink, while others stated that water was better to take with MPs.

In general, children tended to give subjective feedback, as distinct from objective descriptions; their comments ranged from “amazing” through “disgusting” and “yuck”. Yet, the majority of the subjective comments were negative, in contrast to the findings of hedonic scales, which oscillated around neutral responses (the mode of all scores = 3). The reason behind this may be that children tend to prefer familiar or recognisable products [26] and MPs, as a novel product, are difficult to describe. The unfamiliarity of MPs was highlighted by children who were trying to relate them to a more familiar product like “flour” or “sugar”.

Observation of negative facial expression was used as an indicator of sample disliking. The most commonly observed facial expression was “lips pressed”, observed in 37% of administrations, followed by “nose wrinkling” 31%, “brows pulled together and lowered” 23%, “eyes squeezed” 20%, “voice disgust” 11%, and “head shake” 6%. Children showed at least one negative facial expression in 56% of the administrations. Although the largest dose seemed to be causing most discomfort to children, the difference between samples was not statistically significant (Table 3).

### 3.3. Willingness to Take the MPs Every Day as a Medicine

The proportion of participants willing to take the sample every day if they were sick was used as a predictive measure of future and repeated acceptability, which is likely to influence adherence. The smallest dose was most acceptable among children. Based on their responses, 69.4% (27/39) of children would be willing to take the 0.5-mL dose of MPs every day if they were sick, in contrast to 58.8% and 51.7% for 1.2-mL and 2.0-mL doses, respectively (Table 3). In relation to the findings based on hedonic perception, the preference for the 0.5-mL dose can be contributed to by its small volume, rather than the taste or mouthfeel.

A positive association was found between the willingness to take the dose every day as a medicine and ratings on the hedonic scales for all four parameters (Wilcoxon rank test, *p* > 0.05). For example, when the mouthfeel of the sample was acceptable (score ≥ 3), the child would tend to be willing to take the dose every day as a medicine.

### 3.4. Water Consumed and Residual Multiparticulates in the Mouth

Participants had free access to drinking water to aid sample swallowing, no suggestion on the amount of water to take was given. Children consumed 53 mL of water on average (median = 43 mL, min = 1 mL, max = 136 mL) (Figure 4A), in line with findings of Lopez et al. (2018b). The volume of water consumed was comparable between formulations (Wilcoxon test, *p* > 0.017). The volume of a drink of choice taken with the sample was not measured.

After administration of the sample and consuming their preferred amount of water, children were asked whether they could still feel any MPs in their mouth (self-reported). Overall, children reported the feeling of residual MPs in the mouth on 52 occasions (39%). Reported perception of MPs in the mouth increased with larger sample volume, even though this trend was not statistically significant (Chi2, *p* > 0.05) (Figure 4B).

Participants’ spontaneous descriptions of the samples reflected the necessity of a large amount of water to clean MPs residue from the mouth. Children described it as: “I had to take half the water out of the cup to wash it down” or “needed to drink”. Additionally, the researcher’s comments mentioned “lots of swishing of water to help swallow”.

### 3.5. Impact of Age

An association was found between children’s age and the proportion of each sample that was completely swallowed, as opposed to partially swallowed, spat out or refused (Kruskal–Wallis, *p* < 0.05); no such association was found for willingness to take MPs if a child were sick (Kruskal–Wallis, *p* = 0.779) (Figure 5). When the ratings given for hedonic scores (grittiness, volume, taste and mouthfeel) were compared, there was no difference found between age groups (Kruskal-Wallis, *p* > 0.05), suggesting the same level of liking of the sample in the whole study population.

Children tended to choose responses at the middle and extreme ends of the scale as if they were responding on a three-point scale (binominal test, *p* < 0.001). Although such behaviour is typical for younger children (5–6 years old) [27], here, it was observed across all age groups (Kruskal–Wallis, *p* = 0.989).

### 3.6. Method of Administration

Children who expressed a wish to self-administer doses of MPs were allowed to do so. Of nine children who self-administered the doses, there were three who were 10 years old, one who was 9 years old, three who were 8 years old, one who was 6 years old and one who was 5 years old. They were more likely to completely swallow the sample, as compared with children who were given the dose by parent/carer (Chi2 test, *p* < 0.01); no such association was found for the willingness to take the dose every day as a medicine (Chi2 test, *p* = 0.514) (Figure 5). Moreover, these children tended to give more positive ratings on the volume hedonic scale than those who were given the dose by a parent/carer (Mann–Whitney test, *p* < 0.05), no such association was found for other hedonic measures.

Based on the researcher’s comments and photographic documentation of the study, we observed several dosing techniques used by parents/children (Figure 6). Concerning the oral syringe positioning, some parents/children hover the oral syringe above the open mouth during dosing, while others do so above the child’s tongue sticking out of the mouth. Some children hold the oral syringe with lips or teeth. There were also instances of children blocking the oral syringe opening with their tongue, which effectively stopped the flow of MPs. A video of a parent administering a dose to a child is available in the Appendix A.

## 4. Discussion

### 4.1. Effect of Dose Volume on Acceptability and Palatability

The increasing volume of administered MPs negatively impacted the success of taking the sample and the child’s willingness to take a dose. These data suggest that the maximum dose of MPs that can be comfortably taken in a single mouthful by children may be limited; in our study acceptability reduced from 87.5% in terms of success in swallowing for the 0.5 mL (0.32 g) dose to 55% for the 1.2 mL dose (0.78 g). This is of particular relevance for children who have a small oral cavity and lower saliva volume compared with adults (Watanabe et al., 2017). Similar findings of smaller dose volumes being preferred were previously reported in adults (Lopez et al., 2016).

Based on participant hedonic ratings of the sample volume children showed a clear preference for the smallest dose, in line with other acceptability measures and anecdotal comments from participants. The sample grittiness was a dominant negative attribute which could be a cause of participants discomfort and sample dislike, as demonstrated by negative facial expressions and spontaneous feedback. Perception of a “gritty” feeling in the mouth produced by MPs was previously reported and related to the MPs size and amount (Kimura et al., 2015; Lopez et al., 2016). Besides, if the volume of powder or granulate is too large to be suspended in saliva, it can cause a dry and abrasive feeling (Mouritsen et al., 2017). The human mouth can perceive grittiness of particles as small as 6–10 µm (Imai et al., 1995), therefore, a certain level of grittiness of MPs is unavoidable.

The study showed that the small volumes of MPs can be well accepted “as it is”, meaning without the need for a co-administration vehicle or adding sweeteners, flavourings. This is of importance, as in order to ease administration and improve the palatability of sprinkle dosage forms, mixing with food has been common practice (Food and Drug Administration, 2018b; World Health Organization, 2010). Several studies have shown, that administration of MPs in a viscous vehicle can mitigate the feeling of grittiness (Imai et al., 1995; Lopez et al., 2016; Lopez et al., 2018a). However, despite the potential to improve palatability, this additional step before administration increases the burden on the parent/carer and may actually decrease overall acceptability of the product (MacDonald et al., 2006). In addition, a necessity to mix MPs with a vehicle prior to administration would eliminate the benefit of dosing directly to the mouth offered by an oral syringe.

### 4.2. Novelty of the Device

The majority of children (87.5%) were familiar with oral syringes, yet administration of a different type of formulation (MPs) in an oral syringe was a new experience for both the children and their parents/caregivers. Usually, the novelty has a negative impact on one’s liking (Sondergaard and Edelenbo, 2008). It needs to be noted that the data presented are based on the experience and opinions of healthy participants after administration in a single study visit ranging in age from 4 to 12 years. This single-day experience may not accurately represent long-term use, as the acceptability could change over time. On one hand familiarity and liking can increase with exposure to a novel product (Stolzenbach et al., 2013), on the other hand first negative impression can last over a long time. Nevertheless, previous research suggests that training, as well as positive attitudes towards taking a medication or a therapeutic regime, are good predictors of patient compliance (Ghuman et al., 2004; Kreivi et al., 2014).

The results from this study were compared to a previous study where MPs dispersed in 3 mL water were administered from a medicine spoon. In the study of Lopez et al. (2018b), a dose of 0.5 g MPs was administered; the doses administered from the oral syringe were 0.32 g (0.5 mL), 0.78 g (1.2 mL) and 1.29 g (2 mL); the particles used in both studies were of the same size and with the same coating enabling a direct comparison. The acceptability of MPs, in terms of willingness to take a sample every day if a child were sick, was much higher (64% of doses (from pooling data from two smaller doses 0.5 mL and 1.2 mL taken with water)) when dosed with an oral syringe compared to dosing on a spoon (17%). However, opposing results were found when the success of taking a dose was compared between a spoon and an oral syringe (89% vs. 71% completely swallowed doses, respectively). A higher success of taking a dose with a spoon could be attributed to a difference in administration mode. While a dose on a spoon was given in “one go”, administration with an oral syringe allowed to stop and resume dosing at any time during administration which resulted in some doses being only partially administered/swallowed. On the other hand, a higher willingness to take a dose with an oral syringe highlights the importance of the appropriate choice of administration device.

### 4.3. Children Having an Active Role in Taking Medicine

In a paediatric population, the attitude towards medicines and having an active role in taking them are important for adherence (Sanz, 2003). In this study, the nine children decided to self-administer the dose of MPs (with the parental supervision). The results suggest that self-administration increased the probability of the dose being completely swallowed and reduced the negative effect of a large volume. By self-administration, children could adjust the position and speed of dosing to their liking. It is known that the engagement of a child benefits the healthcare process and children who have basic knowledge about medicine related topics can better engage with the process of taking medications (Sanz, 2003). However, children usually have scarce knowledge and negative attitudes towards medicines (Hämeen-Anttila et al., 2006; Syofyan et al., 2019), which can pose a barrier to medicines administration. In this light, the population who took part in the study might not be fully representative of the general population. The participants had to verbally assent to take part in the study which implies at least neutral attitudes to medicines.

### 4.4. Study Limitations and Future Work

In this study, the participants were only provided with written instruction and no feedback during the use of oral syringe was given. Consequently, we could not establish whether training on the correct use of syringe affects the acceptability of a dose. For a novel device, precise instruction and training are crucial for the correct use. Lack of information on where to place the syringe in the mouth led to different methods of administration of MPs. Therefore, the determination of the most appropriate syringe location in the mouth for comfortable and successful dosing would be of interest.

A possible further extension of this study would be to explore methods for administration of larger doses, for instance giving the dose in multiple pushes so that a child swallows smaller volume of MPs. This might be a feasible solution due to the presence of the valve system that allows stopping and starting the MPs flow.

## 5. Conclusions

The study demonstrates that the novel oral syringe being developed is an appropriate device for dispensing doses flexibly and administering neutral-tasting multiparticulate formulations directly to the mouth without the need for a co-administration vehicle. Based on acceptability in a paediatric population, the smallest dose was easiest to completely swallow. The average participants’ hedonic ratings given for taste and mouthfeel were neutral, which highlights that the formulation with no additional sweeteners or flavourings can be highly acceptable in children when dosed directly to the mouth. Additionally, a drink of choice showed no improvement in swallowability when considering higher dose volumes. Self-administration was also noted to have a positive impact on the overall dosing experience when studying larger doses.

## Figures and Tables

**Figure 1 pharmaceutics-12-00806-f001:**
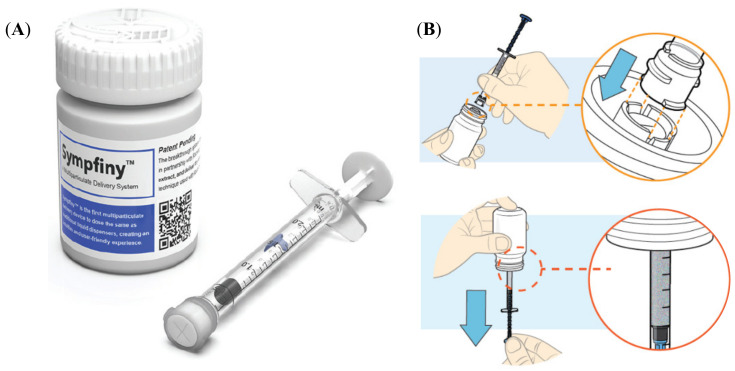
(**A**) Sympfiny™ oral syringe and bottle adapter combination; (**B**) scheme of oral syringe use; top: connecting a syringe to the bottle adapter, bottom: dispensing a dose (reproduced with permission of HS Design).

**Figure 2 pharmaceutics-12-00806-f002:**
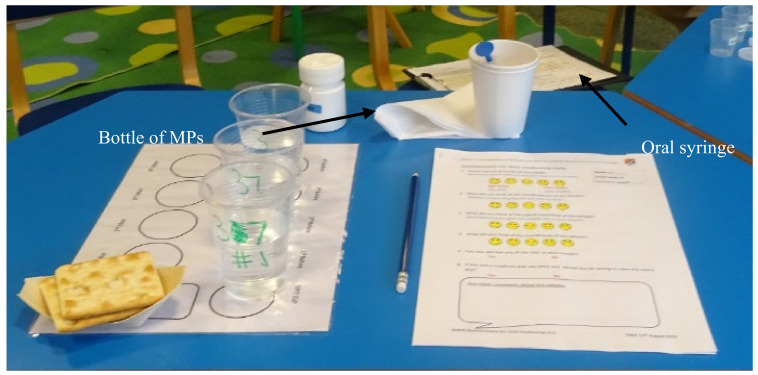
Table set up for one child for the assessment of the acceptability of MPs administered using an oral syringe (Sympfiny™).

**Figure 3 pharmaceutics-12-00806-f003:**
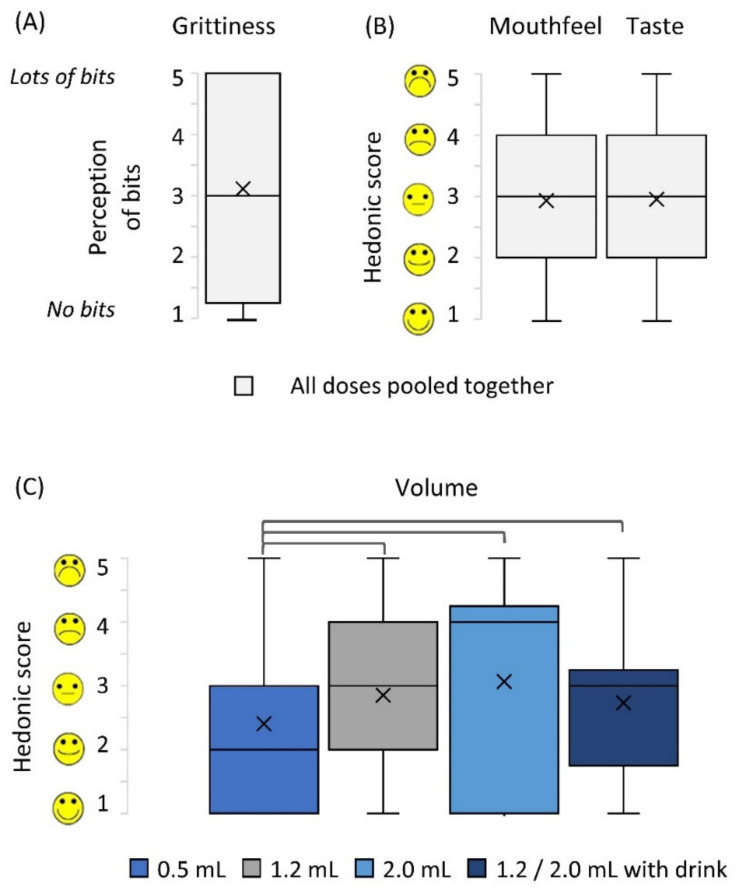
Participant ratings of sample attributes: (**A**) grittiness, (**B**) mouthfeel and taste, (**C**) volume. Parts (**A**) and (**B**) show data pooled from all doses administered as there was no significant difference between doses (Wilcoxon test, after Bonferroni correction, *p* > 0.008); (**C**) shows data per dose, brackets represent statistically different doses (Wilcoxon test, after Bonferroni correction, *p* < 0.008). Centre lines represent the medians, box indicates the 25th and 75th percentiles, whiskers indicate spread of data, *X* represents mean.

**Figure 4 pharmaceutics-12-00806-f004:**
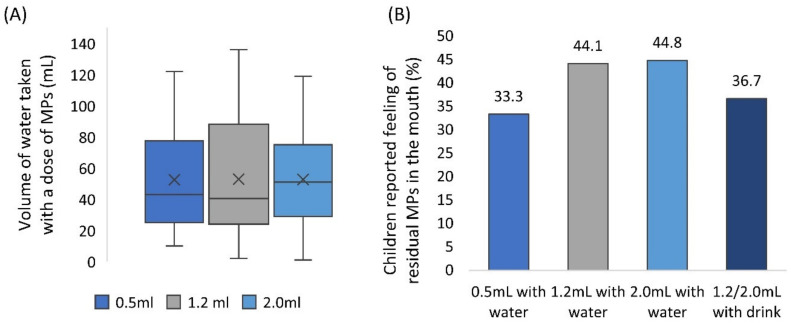
(**A**) Volume of water consumed as a function of a volume of multiparticulates (MPs) dose. Centre lines represent the medians, box indicates the 25th and 75th percentiles, whiskers indicate spread of data, *X* represents mean. (**B**) The proportion of children that reported they could still feel residual MPs in their mouth after sample intake, as a function of a volume of MPs dose.

**Figure 5 pharmaceutics-12-00806-f005:**
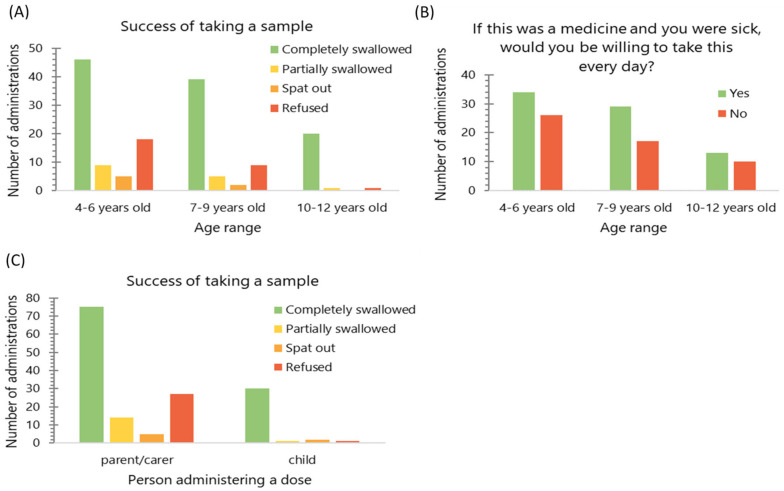
(**A**) Frequency of the dose being completely swallowed (*n* = 156) and (**B**) the number of observations when the child reported willingness to take a dose when sick (*n* = 129) by age group; (**C**) frequency of the dose being completely swallowed (*n* = 156) by the person administering the dose. The data were pooled from all doses administered.

**Figure 6 pharmaceutics-12-00806-f006:**
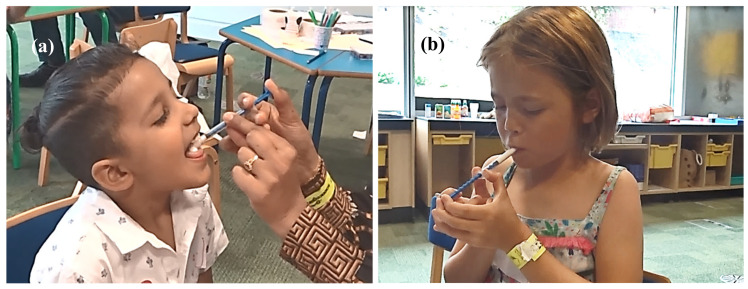
Administration of a dose of MPs using an oral syringe Sympfiny™ by the parent (**a**) and by the participant themselves (**b**) (pictures obtained and reproduced with written parental consent).

**Table 1 pharmaceutics-12-00806-t001:** Tools used by children to assess their experience of swallowing the multiparticulate formulation.

Questions	Assessment Tool
Please rate the grittiness of the sample. (Grittiness means that you can feel ‘bits’ in the sample)	5-point hedonic scale with a dual scale (not gritty–very gritty) (no bits–lots of bits)
What did you think of the overall volume of the sample? (Volume means the amount you had to take)	5-point hedonic scale (unmarked)
What did you think of the overall mouthfeel of the sample? (Mouthfeel means how the sample felt in your mouth)	5-point hedonic scale (unmarked)
What did you think of the overall taste of the sample?	5-point hedonic scale (unmarked)
Can you still feel any of the “bits” in your mouth?	yes/no
If this was a medicine and you were sick, would you be willing to take this every day?	yes/no
Any other comments about this sample	free text

**Table 2 pharmaceutics-12-00806-t002:** Participants demographics.

	Children (*n* = 40)	
Number of Participants	Frequency	Percent (%)
Gender		
Male	21	52.5
Female	19	47.7
Age (years)		
4	6	15
5	10	25
6	4	10
7	4	10
8	9	22.5
9	1	2.5
10	4	10
11	1	2.5
12	1	2.5
Ethnicity *		
Asian	6	15
British	8	20
Mixed	1	2.5
White	23	57.5
Missing **	2	5
Does the child in your care have experience of taking oral medicines from an oral syringe (e.g., Calpol)?		
Yes	35	87.5
No	2	5.0
Missing **	3	7.5
Has the child in your care previously experienced problems with swallowing medicines?		
Yes	6	15
No	31	77.5
Missing **	3	7.5
If you answered YES to the previous question, what caused problems with swallowing medicines?		
Size	0	0
Taste	6	100
Texture	1	16.7
Aftertaste	1	16.7
Dry mouth	0	0
Other, please give details	0	0

* Free text was permitted for participants to complete ethnicity; ** participant did not answer the question.

**Table 3 pharmaceutics-12-00806-t003:** Acceptability of multiparticulate formulation administered using an oral syringe (Sympfiny™) (*n* = 40 children); the dose volumes 0.5 mL, 1.2 mL and 2 mL, correlated to a mass of MPs of 0.32 g, 0.78 g and 1.29 g, respectively.

Administration Details	0.5 mL	1.2 mL	2 mL	With Drink	Difference between the Doses **
Success in swallowing the formulation					
Completely swallowed	35 (87.5%)	22 (55.0%)	23 (57.5%)	25 (62.5%)	0.001
Partially swallowed	3 (7.5%)	7 (17.5%)	4 (10.0%)	4 (10.0%)	0.433
Chewed on	1 (2.5%)	1 (2.5%)	0 (0.0%)	1 (2.5%)	0.392
Sample spat out	2 (5.0%)	3 (7.5%)	2 (5.0%)	0 (0.0%)	0.392
Sample got stuck in throat	0 (0.0%)	1 (2.5%)	0 (0.0%)	0 (0.0%)	0.392
Refused to take sample	1 (2.5%)	6 (15.0%)	11 (27.5%)	10 (25.0%)	0.001
Sum of negative facial expressions (frequency) *					0.167
0—no discomfort	18 (46.15%)	16 (47.15%)	8 (27.59%)	16 (53.33%)	
1 or more—discomfort	21 (53.85%)	18 (52.85%)	21 (72.41%)	14 (46.67%)	
Willingness to take the sample if a child were sick *					<0.05
Positive willingness	25 (69.44%)	20 (58.82%)	15 (51.72%)	16 (53.33%)	
Negative willingness	11 (30.56%)	14 (41.18%)	14 (48.28%)	14 (46.67%)	

* When children refused a sample, the data were not recorded; results were calculated based on evaluated samples (= total number of samples—refused samples). ** Friedman test, *p* value.

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
