# Peer review of "A Novel Oral Syringe for Dosing and Administration of Multiparticulate Formulations: Acceptability Study in Preschool and School Children"

_pharmaceutics, 2020, doi:10.3390/pharmaceutics12090806_

Round 1

Reviewer 1 Report

I have carefully read this paper, and value the effort as provided and submitted. The methods are appropriate for the research question, as part of the medical device development. 

I only have one suggestion: is there value to add a video (has to be checked by the editorial office on the feasibility within this journal) on the manipulation and perhaps on the study execution to further improve the visualisation of the medical device. 

Author Response

In order to address the suggestions of reviewer #1 we are adding a video, captured during the study delivery, which presents a parent administering a 1.2 mL dose of multiparticulate formulation to a 7-year old. The video is provided in MP4 format to be added as a Supplementary Material.

This is also refered to in the text of the revised manuscript

Link to new video: https://youtu.be/6ncPY_Rw8EY  

I can share this as an MP4 format as required

Reviewer 2 Report

The manuscript presented by Hofmanová and co-workers offers a valuable study on the use of a novel medical device for dispensing multiparticulates to pediatric patients. The paper is well written and the experiments well designed. Results are clearly presented and discussed.

I have only a comment (curiosity):

In the section "4.2 Novelty of the device" the authors wrote:

The acceptability of MPs, in terms of willingness to take a sample every day if a child were sick, was much higher (64% of doses, (from pooling data from two smaller doses 0.5 mL and 1.2 mL taken with water)) when dosed with an oral syringe compared to dosing on a spoon (17%). However, opposing results were found when the success of taking a dose was compared between a spoon and an oral syringe (89% vs 71% completely swallowed doses) 

How could the authors justify those opposing results? Please, explain/add a short discussion/hypothesis.

Author Response

In order to address the suggestions of reviewer #2 we extended the discussion to provide our justification for the difference in the acceptability of multiparticulate formulation administered with an oral syringe or a spoon. We hypothesise that the mode of administration impacts the success of taking a dose. While a dose on a spoon was given in “one go”, administration with an oral syringe allowed to stop dosing at any time which resulted in some doses being only partially swallowed. On the other hand, the analysis of the data suggest that the choice of a device is important for the child’s willingness to take the sample as a daily medicine.

This is available in a tracked changes version of hte manuscript attached.
